# Suppressing torsional buckling in auxetic meta-shells

Aref Ghorbani ®[1] ✉, Mohammad J. Mirzaali ®[2], Tobias Roebroek ®[1], Corentin Coulais[3], Daniel Bonn ®[3], Erik van der Linden[1] & Mehdi Habibi ®[1] ✉

Take a thin cylindrical shell and twist it; it will buckle immediately. Such unavoidable torsional buckling can lead to systemic failure, for example by disrupting the blood flow through arteries. In this study, we prevent this torsional buckling instability using a combination of auxeticity and orthotropy in cylindrical metamaterial shells with a holey pattern. When the principal axes of the orthotropic meta-shell are relatively aligned with that of the compressive component of the applied stress during twisting, the meta-shell uniformly shrinks in the radial direction as a result of a local buckling instability. This shrinkage coincides with a softening-stiffening transition that leads to ordered stacking of unit cells along the compressive component of the applied stress. These transitions due to local instabilities circumvent the usual torsional instability even under a large twist angle. This study highlights the potential of tailoring anisotropy and programming instabilities in metamaterials, with potential applications in designing mechanical elements for soft robotics, biomechanics or fluidics. As an example of such applications, we demonstrate soft torsional compressor for generating pulsatile flows through a torsion release mechanism.

Compressing a cylindrical shell along its long axis beyond a certain threshold leads to an often unwanted structural failure known as buckling. The abundance of cylindrical shells in natural and artificial systems such as veins, silos, and cans, has motivated many efforts to predict their buckling behavior and stability landscape under compression[1–4].

In addition to axial compression, cylindrical systems are often subject to torsional loads that can also lead to structural failure, known as torsional buckling[5–8]. Torsional buckling is abundant in nature and everyday life, and can for example threaten the blood flow through arteries[9–11] or damage aircraft wings. Twisting an empty beverage can is a familiar example of torsional buckling that leads to the emergence of creases in the shell (Fig. 1a)[5,12]. A thin cylindrical shell made of rubber buckles similarly, involving multiple creases. Preventing the buckling of soft materials is usually challenging. In the case of a rubber shell, like a garden hose, even increasing the shell thickness does not necessarily

prevent its buckling under torsion but changes the buckled shell's shape, with collapsed cross-section in the middle of the cylinder (Fig. 1b).

Here, we aim to explore the possibility of controlling or even preventing buckling by using metamaterials. Mechanical metamaterials often exploit local instabilities to induce unusual functionalities and exotic mechanical properties such as negative Poisson's ratio in auxetic metamaterials[13–15]. We introduce a strategy to employ local instabilities in metamaterials in order to suppress the torsional buckling instability.

Through experiments and Finite Element (FE) simulations, we demonstrate that auxetic cylindrical shells with periodicity along helical paths are able to prevent torsional buckling, exhibiting radial contraction upon torsion (Fig. 1c). The anisotropy in these meta-shells is associated with a nonmonotonic axial strain that changes from positive under low torsion to negative in case of a large torsional deformation, representing a sign-switching Poynting effect. In

[1]Laboratory of Physics and Physical Chemistry of Foods, Wageningen University, 6708 WG Wageningen, The Netherlands. [2]Department of Biomechanical Engineering, Delft University of Technology, 2628 CD Delft, The Netherlands. [3]Institute of Physics, University of Amsterdam, 1098 XH Amsterdam, The Netherlands. ✉e-mail: aref.ghorbani@wur.nl; mehdi.habibi@wur.nl

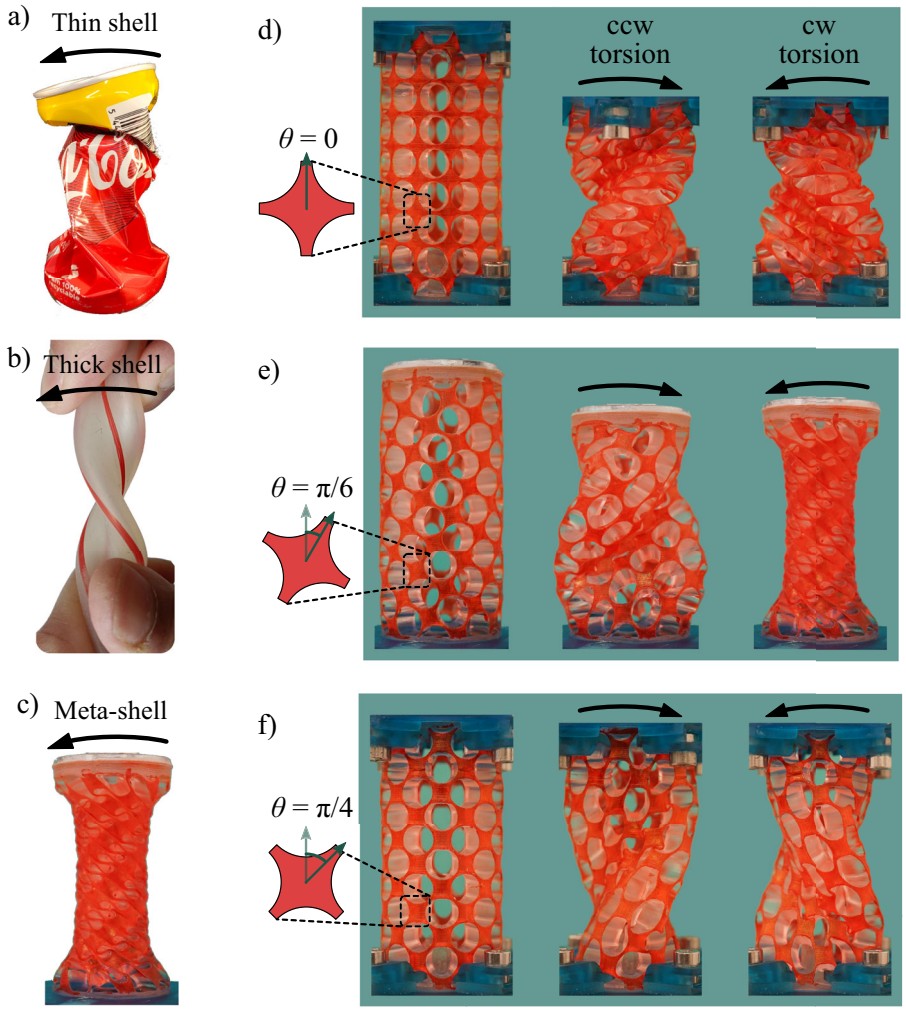

**Fig. 1 | Buckling versus radial contraction in cylindrical shells.** Under counter-clockwise (ccw) and clockwise (cw) torsion, a very thin shell, like a beverage can (**a**), is shrunk and randomly buckled, while a thicker shell (**b**) shows both buckling and radial contraction. In contrast, a designed meta-shell (**c**) shows a more uniform radial contraction under torsion instead of buckling. Varying the orientations of the meta-shell unit cells results in different types of deformation under torsion (**d–f**). The helical meta-shell radially contracts under cw torsion (**e**, right).

isotropic systems, shear deformation usually leads to perpendicular dilation (Poynting effect[16]) or to contraction (reversed Poynting effect[17]). However, an anisotropic system can exhibit a sign-switching Poynting response[18], where shear-induced dilation and contraction can be realized in different deformation ranges. We introduce the negative Poisson's ratio and anisotropy associated with sign-switching Poynting response as two essential ingredients to design cylindrical shells resistant to torsional buckling.

We use the well-known auxetic holey sheet[19,20] to design our cylindrical metamaterials (meta-shells). The holey sheet consists of a 2D array of closely packed circular voids in an elastic sheet. The negative Poisson's ratio of such holey sheets is due to the distinct reconfiguration of their unit cells upon uniaxial loading along their principal axis[21] that coincides with a nonlinear mechanical response[22,23]. Holey sheet patterns have in the past inspired shape-transforming 2D auxetic metamaterials with thermally tunable auxeticity[24], hierarchical folding[25], and global bistability[26]. Additionally, the holey sheet pattern has been used to design uniformly collapsible spherical shells when actuated by reducing the pressure inside the shell[27]. Holey sheet patterns can also be exploited for engineering cylindrical shells that display auxeticity upon axial loading[28,29] or deliver bending or twisting upon actuation by reducing the inner pressure[30]. Various deformation modes and auxeticity can be achieved

by employing local rotations in soft network structures[31], which can give rise to a global twist via stretching in cylindrical systems[32–34]. More in general, the coupling between torsion and compression and the sign of the Poynting effect is programmable in non-auxetic cylindrical metamaterials[35].

Previous studies of auxetic structures focused on mechanical properties under uniaxial loading in a relatively small deformation range where an out-of-plane buckling is forbidden. Some studies unraveled novel global buckling behaviors such as discontinuous buckling of auxetic beams[36] and porosity-depended buckling in holey cylindrical shells[28]. Here, we focus on preventing torsional buckling in auxetic meta-shells subjected to large torsional deformations.

We tune the orthotropy orientation of the meta-shells by rotating the principal axes of the network grid, and consequently all the unit cells with respect to the main axis of the cylinder. By rotating the principal axis, we design cylindrical meta-shells whose unit cells are asymmetric with respect to the vertical axis, and periodic along helical paths. We find that these helical meta-shells behave differently depending on the torsional direction. Twisting a helical meta-shell triggers a unique structural reconfiguration with a significant uniform negative radial strain (radial contraction) and nonmonotonic axial strain. The structural reconfiguration occurs via a snap-through softening followed by stiffening due to the ordered stacking of unit cells

that coincides with a radial contraction and that prevents torsional buckling. The absence of torsional buckling and torsion-tunable cylinder radii are unique features exploitable in soft robotics, bio-mechanics, and material engineering for designing functional soft systems such as pumps, valves, and actuators. Here, we explore auxetic cylindrical shells for designing a cylinder with localized contraction under torsion and a soft pulsatile compressor.

## Results and discussion

### System definition

A square arrangement of four circular voids in a 2D sheet shaped into a cylinder can be described by a square-like unit cell along two principal axes, as shown in Fig. 1d, left. The mechanical properties of the system can be altered by rotating the orientation of the principal axes with respect to the vertical axis over an angle $\theta$. Setting $\theta = 0$ aligns one principal axis with the cylinder's vertical axis (Fig. 1d, left). The mechanical properties of this 2D system have been studied compre-hensively. By varying the void shape from circular to elliptical shapes, it was shown to be an orthotropic system[22]. We find that a cylindrical meta-shell based on such a unit-cell arrangement behaves similarly under clockwise (cw) or counterclockwise (ccw) torsions since it is symmetric concerning the torsional deformation around the $z$-axis of the shell (Fig. 1d).

We rotate the principal axes of the auxetic lattice, and therefore the orthotropy orientation, with respect to the loading direction (torsion around the main axis of the cylinder) to create meta-shells that behave differently depending on the direction of the torsional (shear) deformation. If the rotation angle $\theta$ is between 0 and $\pi/4$ (or between $\pi/4 <$ and $\pi$), the mirror symmetry of the unit cells with respect to the vertical axis is broken. As a result, initially horizontal and vertical lines that connect the unit cells become helices with opposite handedness (growing with opposite rotations) and different pitch values on the cylindrical shell (Fig. 1e, left). Therefore, we refer to these shells as helical meta-shells. Similar helical geometries have been shown to provide a flexible platform for designing soft robot arms that can exhibit various modes of deformation and perform complex tasks[37].

Periodicity along helical paths reflects a global chirality in the cylindrical structure. A system is chiral if it cannot be mapped to its image under parity inversion (mirroring) via any transition and/or rotation. Chirality in metamaterials is often implemented locally in the unit-cell design[38-42]. Even though the unit cells in our systems are not chiral, the meta-shells that break symmetry exhibit global chirality. The mirror image of our meta-shells whose unit cells are rotated by $\theta$ is equivalent to a meta-shell created by a $-\theta$ rotation, which cannot be superimposed onto each other. However, the meta-shell created by $\theta = \pi/4$ (Fig. 1f, left) is not chiral as the helical paths connecting the unit cells along the principal axes have the opposite handedness but the same pitch values. Consequently, the behavior of this meta-shell remains insensitive to the direction of torsion, and we do not classify it as a helical meta-shell. Our results below highlight the significant dif-ferences between the helical and non-helical meta-shells in their mechanical characteristics. Our meta-shells have initial height of $h_0 = 53.3$ mm and inner and outer radii of $R_{0,min} = 7.5$ mm and $R_{0,max} = 12.5$ mm, respectively. Thus, the shell thickness is 5 mm for all meta-shells. The critical shell thickness, below which the meta-shells become incapable of demonstrating the desired behavior is discussed in the Supplementary Information (SI, Critical shell thickness). Further details regarding the meta-shells' design and fabrication process can be found in the Methods section.

We investigate the meta-shells experimentally and using compu-tational modeling performed by finite element (FE) simulations. The simulations are described in the Methods section. To experimentally study the properties of our meta-shells, we apply clockwise (cw) and counterclockwise (ccw) torsion on each cylinder. During torsion, the cylinders are axially free (axial force is $0 \pm 0.1$ N), which means that the

cylinders can freely dilate or contract depending on their normal force response. Similarly, a soda can is free to contract while manually twisting it, and becomes shorter due to the application of torsion (Fig. 1a). The applied torsion and torque around the main axis of the meta-shells are designated by $\varphi$ and $\tau$, respectively. We calculate the axial strain induced by torsion by $\delta_n = |h - h_0|/h_0$, where $h$ is the height after the applied compression. This strain should not be confused with the applied compression strain, $\delta$. The compression stress is given by $\sigma = F/A_s$, where $F$ is the compression force, and $A_s = \pi(R_{max}^2 - R_{min}^2)$ is the area of a horizontal cross-section of the cylindrical shell. Since the thickness of our meta-shells is significantly smaller than their height and perimeter, twisting them is assumed to be equivalent to shearing a thin plate. Therefore, the shear strain is defined as $\gamma = \varphi R/h_0$, and the shear force is represented by $F_s = \tau/R$, where $R = (R_{max} + R_{min})/2$ is the average radius. Furthermore, the radial strain of the cylinders is defined by $e_r = (r - R_{max})/R_{max}$, where $r$ is the outer radius of the deformed shells at height $z = h/2$. Response of the meta-shells under compression is discussed in the SI: Buckling under compression.

### Contraction versus buckling under torsion

We apply cw and ccw torsion on the meta-shells under zero axial load, where the shells can freely dilate or contract in the axial direction during torsion. As can be seen in Fig. 1d–f, we observe various beha-viors depending on the unit-cell rotation angle with respect to the radial direction, $\theta$. In the symmetric structures ($\theta = 0$ and $\pi/4$), we observe torsional buckling although upon different critical torsional angles, as shown in Fig. 1d and f for cw and ccw torsion, respectively. It can also be seen that, since the designs are symmetric, cw torsion leads to the same buckling behavior as ccw torsion, but with an inverse twist. For helical meta-shells with $0 < \theta < \pi/4$, the deformation strongly depends on the direction of applied torsion, as exemplified by Fig. 1e, left, where $\theta = 31° \approx \pi/6$. In case of ccw torsion, buckling is inevitable (Fig. 1e, middle), while under cw torsion, this is not the case, and the meta-shell uniformly contracts in the radial direction (Fig. 1e, right). The origin of this radial contraction (negative radial strain) is rooted in the auxeticity of the structure.

### Torsion-induced negative radial strain

Next, for a meta-shell with $\theta = \pi/6$ (Fig. 2a), we investigate the radial contraction as a function of cw torsion angle $\varphi$, exemplified in Fig. 2b–d. We find that the folding mechanism underlying the radial contraction in this scenario is entirely different from that of a structure with $\theta = 0$ under compression.

In the latter case, which is the most familiar one[19], the unit cells on the principal axes along the compression direction fold on each other by oppositely rotating around their out-of-plane axis (corresponding to the radial direction in the cylinder)[19,28]. This is illustrated in the top panels of Fig. 2e, where the unit cells are labeled as the elements of a $2 \times 2$ grid, i.e. *1;1*, *1;2*, *2;1*, and *2;2*. During buckling, unit cell *2;1* folds on *1;1*, and *2;2* folds on *2;1*[19]. However, in the meta-shell with $\theta = \pi/6$ under torsion (bottom panels of Fig. 2e), the unit cells in the diagonal of the grid (*1;1* and *2;2*) fold on each other by rotating in the same direction around their out-of-plane axis. This folding mechanism under torsion leads to a negative radial strain due to compaction and filling of the voids. FE analysis confirms the torsion-induced radial shrinkage. The same stages of deformation of Fig. 2a–c are simulated in Fig. 2f–h, where the color scale represents the local equivalent stress, $S_{eqv}$ (known as von Mises stress). The vertical cross-section of the deformed meta-shell (Fig. 2h, right) reveals the symmetry of the deformation along the meta-shell. The horizontal cross-sections of the meta-shell of height $h$, taken in the middle ($z = h/2$), illustrate the azimuthal symmetry and uniformity of the contraction (Fig. 2i).

To further investigate the contraction, we quantitatively study the radial strain as a function of cw torsion. In our experiments, we only consider the helical meta-shells ($\theta \simeq \pi/12$, $\pi/6$). In our FE simulations,

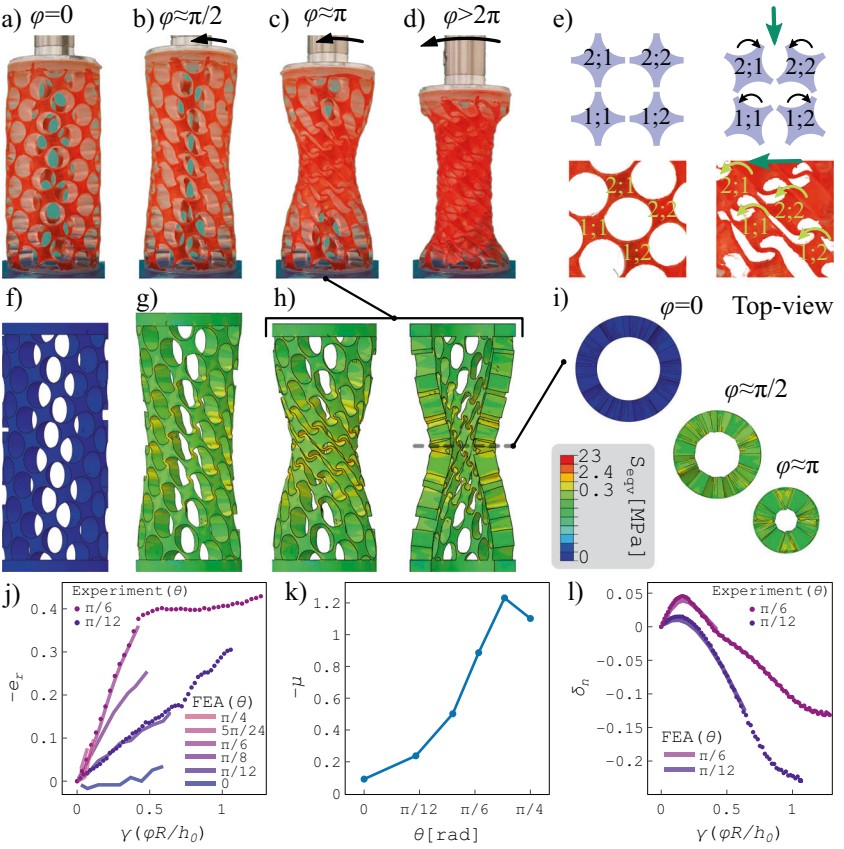

**Fig. 2 | Negative radial strain under torsion. a–d** A helical meta-shell with $\theta = \pi/6$ is deformed at increasing cw torsion angles. **e** Non-sheared (left) and sheared (right) unit-cell configuration, for a meta-shell with $\theta = 0$ (top) and $\theta = \pi/6$ (bottom). Unit cells are labeled as 1;1, 1;2, 2;1, and 2;2, as they are part of a 2 × 2 grid. **f–h** FE simulations of the deformation of a helical meta-shell with $\theta = \pi/6$ upon cw torsion, where the color scale indicates the equivalent stress ($S_{eqv}$). Image (h, right) displays the vertical cross-section of the deformed meta-shell. **i** Horizontal cross-sections of the meta-shell in different stages of deformation. **j** Induced negative radial strain as a function of the shear strain for meta-shells with different unit-cell orientations. Circles indicate the experimental values and solid lines represent the FE results. **k** The negative of the contraction ratio as a function of unit-cell orientation. **l** Nonmonotonic axial strain of the helical meta-shells with $\theta = \pi/12$ and $\theta = \pi/6$, as a function of the applied shear strain.

we include additional meta-shells ($\theta \simeq 0$, $\pi/12$, $\pi/8$, $\pi/6$, $5\pi/24$, $\pi/4$) to obtain a comprehensive understanding of the phenomenon. The values of $\theta$ are approximately integer multiplies of $\pi/24$, and the exact values are reported in the Methods section. For simplicity, we refer to the approximate values.

In Fig. 2j, we show the negative of the radial strain, $-e_r$, or radial contraction as a function of the shear strain, $\gamma = \phi R/h_0$ for different values of $\theta$. The radial contraction in the middle of the meta-shell increases linearly as a function of the applied shear strain for $\gamma < 0.4$. We define the slope of $e_r$ versus $\gamma$ as the shear-induced contraction ratio $\mu$, which is constant in the low shear strain regime ($\gamma < 0.4$) and depends on $\theta$. Its magnitude is comparable to the Poisson's ratio, $\nu$, which is defined as the ratio of the induced lateral strain to the applied axial strain. The minimum Poisson's ratio in our designed meta-shells, representing the compression-induced contraction of the shell, is obtained for the meta-shell with $\theta = 0$ as $\nu \simeq -0.5$, in agreement with previous studies[19,43]. Surprisingly, the shear-induced contraction is significantly more prominent for the meta-shell with $\theta \geq \pi/6$, where $\mu < -0.9$. This is shown in Fig. 2k, where we plot $\mu$ versus $\theta$ based on the FE results. Even though the contraction ratio is higher for meta-shells with $\theta = 5\pi/24$ and $\pi/4$, their contraction is small ($|e_r| < 0.1$) as they only remain stable at the low-shear limit ($\gamma < 0.1$).

The origin of this substantial difference between the compression- and shear-induced contraction, respectively for the meta-shells with $\theta = 0$ and helical meta-shells, is the difference in folding mechanism of the unit cells (Fig. 2e). The folding mechanism under

shearing (torsion) in the helical meta-shells allows higher compaction of the unit-cells upon torsion.

The radial contraction as observed in experiments reaches a plateau where the radius remains almost constant at $r \simeq 0.6 R_{max}$ (Fig. 2c,d). This occurs due to the extreme compaction of the structure, resulting in the deformed unit cells filling the voids and touching the neighboring unit-cells, sharing a large contact area. This level of compaction first emerges around $z = h/2$ (Fig. 2c), and with increasing torsion, it symmetrically extends towards the top and bottom boundaries of the cylinders (Fig. 2d). For the structure with $\theta = \pi/12$, we do not reach the compaction regime (2f). Under extreme torsion, the cylinders become highly squeezed (Fig. 2d) but still resist buckling (see supplementary Video 1). Note that the FE simulations only reproduce the linear contraction results, showing no plateau as observed in experiments.

## Nonmonotonic axial deformation under torsion and sign switching Poynting response

We next study the axial response to torsion, which is a fundamental feature of nonlinear materials[44,45]. The axial stress perpendicular to the shearing direction is present in all elastic and viscoelastic systems. It is rooted in the elastic nonlinearity of isotropic solids and viscoelastic materials. In isotropic systems, the shear-induced axial stress is usually positive, which leads to the dilation of the material when sheared, known as the Poynting effect[16]. Negative axial stress is also observed in some isotropic biopolymer systems[17,46,47]. However, the Poynting response can be significantly different in anisotropic systems.

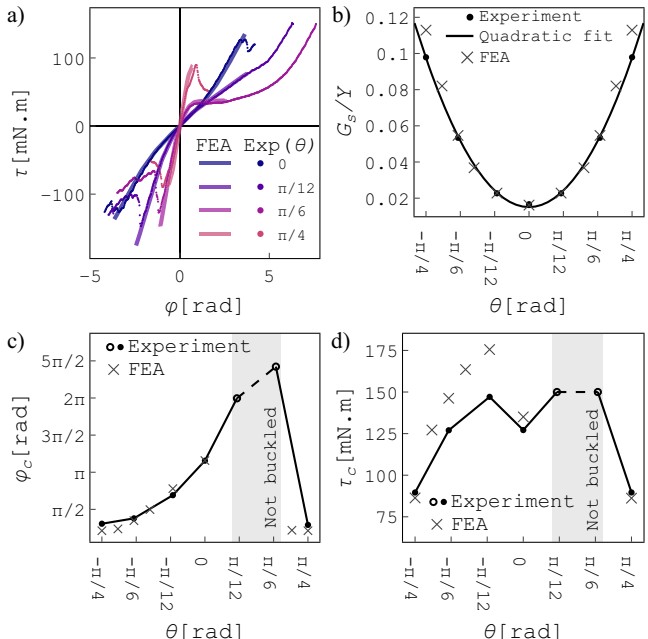

**Fig. 3 | Pre-buckling stiffness and the onset of torsional buckling. a** Torque responses of the meta-shells of varying $\theta$'s as a function of the torsional angle $\varphi$ around the shell axis. Solid lines and circles indicate the FE and experimental results, respectively. **b** Pre-buckling shear modulus $G_s$ rescaled by Young's modulus of the elastomer $Y$, as a function of the unit-cell orientation, $\theta$. Since $G_s$ for $-\theta$ is mirrored with respect to $\theta = 0$, we calculate the values of $G_s$ for the unit cell with opposite rotation using $G_s(-\theta) = G_s(\theta)$. (**c, d**) Buckling torsional angle (**c**) and buckling torque (**d**) as a function of unit-cell orientation upon cw torsion. The buckling values for the meta-shells with $\theta < 0$ are obtained from the ccw torsion experiments since twisting a meta-shell with $-\theta$ unit-cell orientation in the cw direction is equivalent to twisting a meta-shell with $\theta$ unit-cell orientation in the ccw direction. No torsional buckling is observed in the shaded area, but we display the maximum torque values reached during the test by open symbols.

Our helical meta-shells, in addition to the monotonic negative radial strain, exhibit a nonmonotonic axial strain representing the complex Poynting behavior of the system (Fig. 2l). The meta-shells initially show a positive Poynting response by dilating under torsion (Fig. 2b,g). After the axial strain reaches a maximum value ($\delta_n \approx 0.05$), the Poynting response is reversed, and the meta-shells contract and finally become shorter than their initial height (Fig. 2c,d). FE results also reproduce the axial strain of the meta-shells (Fig. 2f–h), in excellent agreement with experiments. The nonmonotonic axial strain is a result of the sign-switching normal stress that emerges due to the particular design of anisotropy associated with the helicity of the meta-shell. Similarly, anisotropic systems with intrinsic helicity have been shown to display a sign-switching axial response[18].

We recently presented a metamaterial in which the sign and magnitude of the Poynting response are programmable via a pre-compression step[35]. In contrast to these materials, our helical meta-shells induce a switch in axial stress from positive at low shear strain to negative at high shear strain. A normal response with a sign reversal transition has also been observed in pantographic structures[48]. On the other hand, the shear-induced negative radial stress is only observed in viscoelastic systems but is usually negligible compared to the axial stress[49]. The radial stress in viscoelastic systems emerges as a result of the coupling between non-linear elasticity and flow. Here, we show the emergence of significant radial stress in purely elastic materials.

It should be noted that the axial strain can only be nonzero if the system is axially free to deform, with the axial force kept zero during torsion. In an alternative scenario, the torsion experiment is performed

under a fixed gap, where the height of the meta-shell is kept constant during torsion. Torsion in this scenario leads to similar radial contraction, which is employed in the "Pulsatile torsional compressor" subsection.

## Small deformations and pre-buckling regime
Tuning the orthotropy via the unit-cell orientation angle $\theta$ highly influences the mechanical responses of the meta-shells under torsion. In Fig. 3a, the torque response is shown as a function of the torsional angle $\varphi$ for different meta-shells with various values of $\theta$. The FE results are initially obtained using a hyperelastic materials model based on Ogden strain energy (see Methods section for details). The results of the FE analysis (solid lines in Fig. 3a) closely reproduce the experimental results (data points).

For small torsional deflections, $\varphi < \pm\,0.2$ rad, all meta-shells are in the pre-buckling regime, and the responses are almost linear. We use the slope of the linear fit of this pre-buckling regime to obtain the shear modulus (shear stiffness) $G_s$, by considering $\tau = G_s J\varphi/h$ where $\tau$ is the torque around the axis of the shell, and $J = \frac{\pi}{2}(R_{max}^4 - R_{min}^4)$ is the second moment of area of the shell[35]. In Fig. 3b, we show $G_s$ as a function of $\theta$. We observe that the shear stiffness is lowest for the meta-shell with a straight unit-cell orientation ($\theta = 0$), and increases to a maximum for the meta-shell with $\theta = \pi/4$. Consequently, setting $\theta = \pi/4$ leads to the highest shear stiffness, where the principal axes of the pattern are aligned with the diagonal directions. Since the unit cells have a rotational symmetry with respect to the axial direction, variation of the shear stiffness must be insensitive to the direction of the unit-cell rotation, i.e. $G_s(\theta) = G_s(-\theta)$. We use this symmetry to include the values of $G_s$ for negative unit-cell rotations ($-\theta$) in the Figure. Additionally, we estimate the shear stiffness using a quadratic fit ($G_s \propto \theta^2$; dashed curve in Fig. 3b). The quadratic relation is verified analytically using a simple tilted beam model (see SI, Shear modulus of a tilted beam). The FE results confirm identical behavior in excellent agreement with the experimental results.

## Large deformation and torsional buckling
In Fig. 3a, we show the torque response of the meta-shells upon cw ($\phi > 0$) and ccw ($\phi < 0$) torsion. By increasing the ccw torsion, the torque response of most meta-shells changes dramatically due to torsional buckling. However, the torque response of the helical meta-shells monotonically increases under a cw torsion, implying that torsional buckling is circumvented in helical meta-shells. As mentioned in the previous paragraph, the meta-shells with $\theta$ and $-\theta$ unit-cell orientations are mirror symmetries around the $z$-axis of the meta-shell. Therefore, twisting our meta-shell with $\theta$ in the ccw direction is equivalent to twisting a meta-shell with a $-\theta$ unit-cell orientation in the cw direction. Thus, we present data under ccw torsions as cw torsion with a $-\theta$ unit-cell orientation to obtain a comprehensive understanding of the effects of unit-cell orientation on the buckling of the meta-shells. In Fig. 3c and d, we show the buckling torsional angle $\varphi_c$ and torque $\tau_c$ of the meta-shells as a function of the unit-cell orientation $\theta$. As helical meta-shells with $\theta = \pi/12$ and $\theta = \pi/6$ do not show buckling under cw torsions, the highest experimental values of $\varphi$ and $\tau$ are shaded in gray in Fig. 3c and d. The experimental limitation is set by the maximum torque that our setup can apply (150 mN m). Within this limit, half of the hinges are aligned and strongly pulled, and two finally break (Supplementary Video 1), but torsional buckling is not observed and the torque value does not drop.

The FE results predict similar values for the onset of buckling in systems that undergo buckling. The buckling torsional angles predicted by FEA are in excellent agreement with the experimental results (Fig. 6c). However, the buckling torques derived from the FE results exhibit a larger deviation from the experimental data (Fig. 6d). Nonetheless, this deviation remains within an acceptable range, with the largest deviation being only 19%, observed for the meta-shell with $-\theta = \pi/12$. This discrepancy primarily stems from inaccuracies in the material model under

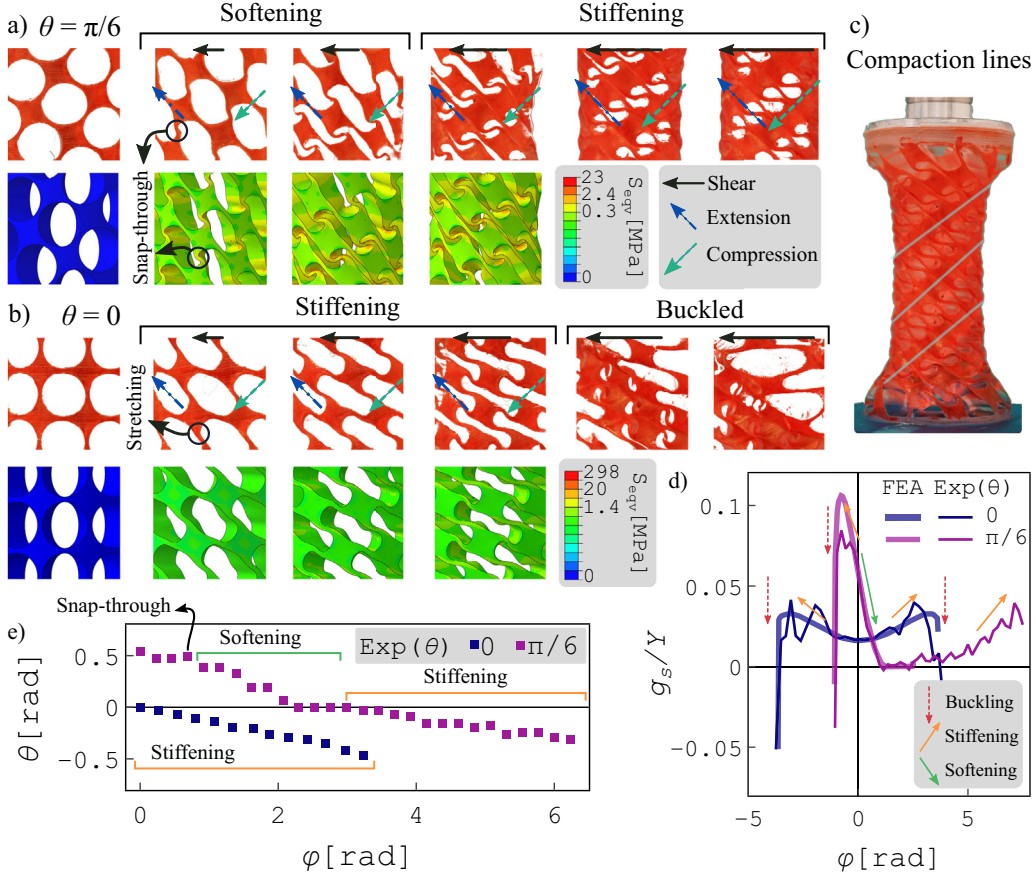

**Fig. 4 | Structural reconfiguration and shear softening-stiffening.**
**a**, **b** Experiments (top) and FE simulations (bottom) of structural reconfiguration under shear deformations for a section of a meta-shell with $\theta = \pi/6$ (**a**) and $\theta = 0$ (**b**). Color scales in FE simulations indicate equivalent stress levels. Arrows points to areas of shear, extension, and compression. **c** Ordered stacking of unit cells in a twisted meta-shell with $\theta = \pi/6$ Solid lines indicate compaction lines. **d** Shear stiffness of the meta-shells as a function of torsion. Arrows indicate buckling, stiffening, and softening events. **e** Evolution of the unit-cell orientation as a function of the applied torsional deflection for meta-shells with $\theta = \pi/6$ (purple) and $\theta = 0$ (blue).

extreme deformation conditions. Moreover, our assumption in defining the contact properties in FE simulations (e.g., friction coefficient of 0.1) may differ from the real system, which could result in deviations from the obtained buckling torque values in the experiments.

In scenarios with no buckling, namely for meta-shells with $\theta = \pi/12$ and $\theta = \pi/6$ under cw torsion (indicated by the shaded area in Fig. 6c and d), FE simulations are unable to reproduce the results due to the complexity of deformation events involving elaborate contact mechanisms. Additional information must be incorporated to predict meta-shells' behavior through FE simulations under such a large deformation range. This includes accurate data on the friction coefficient during self-contact and the incorporation of hyperelastic properties for larger deformations. Moreover, employing elements with high-order interpolation (e.g., brick elements with a higher number of integration points) and a finer mesh may be necessary to capture meta-shells behavior under significant deformation adequately.

**Structural reconfiguration and shear softening-stiffening**
The above results show the potential of the helical meta-shells in preventing torsional buckling even under more than a full turn ($2\pi$) twist (Fig. 3c). Similarly, the energy stored by twisting the cylinders before buckling (or until the maximum torque is reached) depends on the unit-cell orientation and is considerably higher for helical meta-shells with $\theta = \pi/12$ and $\pi/6$ (see SI: Energy perspective). To understand the origin of the torsional buckling circumvention, we investigate the structural reconfiguration and shape-changing of the meta-shells upon torsion.

As the meta-shell is very thin compared to its height and perimeter, twisting it is comparable with shearing a 2D plate. Therefore, the deformation of a small section of the meta-shell can be represented by a shearing, equivalent to the torsion of the whole meta-shell. In Fig. 4a and b, we show the rearrangement of the unit cells under shear deformation for a section of a meta-shell with $\theta = \pi/6$ and $\theta = 0$, respectively. Underneath the experimental images, color-coded images based on FE simulations are presented, with a color scale representing the local equivalent stress. Essentially, shear deformation (full black arrow vector) is a combination of compression (dashed green vector) and extension (dot-dashed blue vector), which are perpendicular to each other.

Under low shear deformations, we observe softening due to snap-through buckling of hinges for the meta-shell with $\theta = \pi/6$, but stiffening due to stretching of hinges for the meta-shell with $\theta = 0$. This difference leads to significant differences in the rearrangement of the two meta-shells' unit cells under large shear deformations. In the meta-shell with $\theta = \pi/6$ (Fig. 4a), the reconfiguration ultimately leads to the stacking of unit cells, roughly aligned with the compression contribution of shear (green vectors), which results in the stiffening of the system. Meanwhile, in this state, half of the hinges are stretched along the extension line (blue vector). This structural reconfiguration coincides with a negative radial strain that results in a cylindrical shell with the same thickness but a smaller radius. In Fig. 4c, we show the squeezed structure under torsion ($\theta = \pi/6$) with the solid lines highlighting the unit-cell stacking direction. Based on these results, we argue that the softening-stiffening transition with a

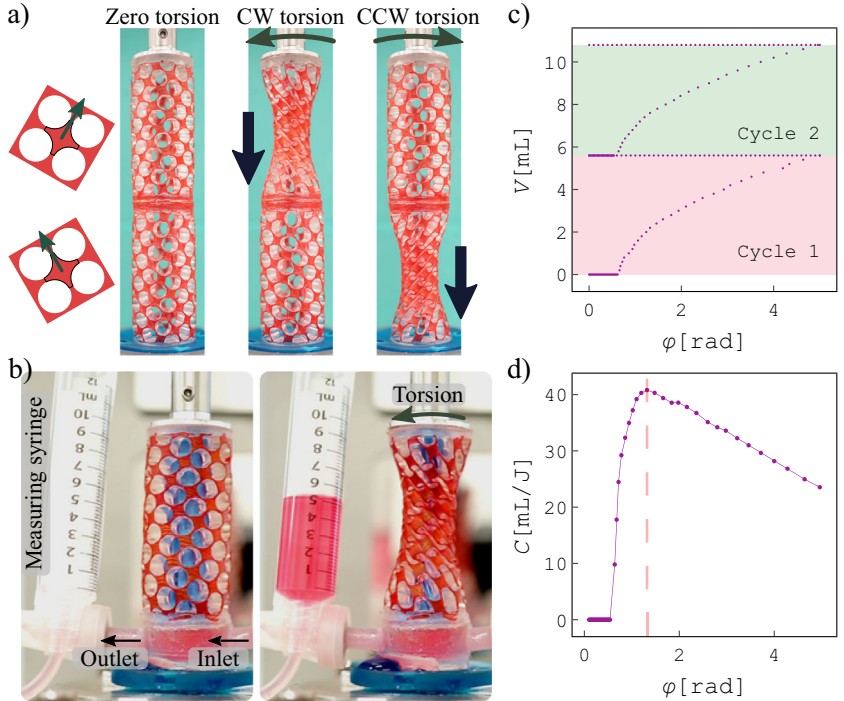

**Fig. 5 | Potential applications. a** Meta-shell consisting of two half-shells of opposite chirality (cross-sections shown on the left) exhibits local contraction in the upper or lower half under clockwise and counterclockwise torsions, respectively, with potential applications in the transportation of yield stress fluids (supplementary Video 2). **b** Demonstration of a torsional compressor used to pump a liquid (colored water) into a measuring syringe. A pulsatile flow can be achieved upon twisting and releasing cycles. The inner surface of the meta-shell is sealed using a thin rubber film (Supplementary Video 3). **c** Pumped volume as a function of torsional deflection over two torsion cycles. **d** Pumping unit capacity $C$, defined as the ratio of the pumped volume to energy input, as a function of torsional deflection $\varphi$.

favorable stacking of unit cells and contraction of the meta-shell increases its stability rather than provoking buckling. However, in the meta-shell with $\theta = 0$ (Fig. 4b), the reconfiguration does not help the stability of the meta-shell, and initial stiffening immediately leads to buckling.

Next, we investigate the softening-stiffening transitions in the mechanical response of the system by plotting the stiffness modulus as a function of torsional deflection in Fig. 4d. The local stiffness is calculated numerically using $g_s = (h/J)(d\tau/d\varphi)$. The meta-shell with $\theta = 0$ stiffens by increasing the shear deformation (orange arrows) and buckles at certain points (red arrows). The helical meta-shell ($\varphi = \pi/6$) similarly stiffens and quickly buckles under ccw torsion. However, under cw torsion, it softens initially (green arrow) and slowly stiffens by increasing the torsion. The FE results closely reproduce the stiffness values for meta-shells as a function of torsion, predicting identical behavior. Fluctuations in the experimental local stiffness values arise from experimental inaccuracies in the torque measurements, which are not relevant for the FE results (see Supplementary Information, Local stiffness calculation). We also show the softening and stiffening regimes upon the structural reconfigurations in Fig. 4a and b, which confirm our visual observations of a snap-through transition and stretching of hinges as the origin of softening and stiffening in meta-shells with $\theta = \pi/6$ and $\theta = 0$, respectively.

In Fig. 4e we demonstrate the evolution of the orientation $\theta$ of a unit-cell, at the middle of the shell (measured manually), as a function of the global torsional angle $\varphi$. Differences between these evolutions provide insights into the differences in local deformations of the meta-shells. For the helical meta-shell (i.e., $\theta = \pi/6$), the initial trend of the unit-cell angle is relatively constant, associated with the snap-through transition at the onset of the softening regime. In the softening regime, initially, it decreases before reaching a relatively constant state towards the end of this phase. Subsequently, during the stiffening phase, which coincides with self-contact and compaction, the unit-cell orientation continues to decrease steadily as a function of the applied torsional angle, but at a lower rate compared to the decrease observed in the softening regime. For the helical meta-shell (i.e., $\theta = \pi/6$), initially, the unit-cell angle exhibits a relatively constant trend, indicative of the snap-through transition at the commencement of the softening regime. Subsequently, it decreases, followed by another period of relatively constant behavior within the softening regime. On the other hand, the unit-cell angle for the meta-shell with $\theta = 0$ steadily decreases as a function of $\varphi$, exhibiting a stiffening behavior until the meta-shell eventually buckles. These observations provide a qualitative description with the distinct structural reconfigurations and morphological transformations of the helical meta-shells as critical factors in preventing torsional buckling.

Here we propose exploiting torsion-induced localized or pulsatile contractions for designing pumps and compressors.

## Localized radial contraction

Based on our observations of coupling between shear-induced negative radial strain and orthotropy orientation in meta-shells, we next design meta-shells with varying orthotropy that exhibit localized deformations under torsion. We create a cylindrical metamaterial, shown in Fig. 5a, left, using two half-length meta-shells with opposite unit-cell orientations ($\theta = +\pi/6$ and $\theta = -\pi/6$), to obtain opposite chirality. These two meta-shells are mirror images and cannot be mapped, indicating the chirality of the system. Twisting in the cw direction leads to the radial contraction of the top half (Fig. 5a, middle), while twisting in the ccw direction induces a radial contraction of the bottom half of the meta-shell (Fig. 5a, right); see supplementary Video 2. This meta-shell can be used as a transportation or locomotion platform for soft robotic applications or a soft compartment to control and create flow.

## Pulsatile torsional compressor

A single helical meta-shell can also be used as a torsional compressor to create a pulsatile flow, as demonstrated in Fig. 5b. We use a meta-shell with $\theta = \pi/6$, whose voids are sealed by a rubber membrane glued to the inner surface of the meta-shell, and equipped with one-way inlet and outlet gates. The outlet is connected to an open syringe used for measuring the amount of pumped liquid (colored water), $V$. This parameter increases nonlinearly as a function of the torsional deflection, shown in Fig. 5c for two cycles; also see supplementary Video 3. Here, the cylinder height is kept constant during torsion.

To determine the optimal torsional deflection amplitude for pumping, we define the unit capacity $C$ as the pumped volume per unit of energy required for twisting. The unit capacity as a function of the applied torsional deflection, presented in Fig. 5d, shows a clear maximum at $\phi = 1.3$ rad, after which it decreases linearly with increasing torsion. The maximum unit capacity provides an indication of the amplitude of torsion in each cycle needed to obtain the highest efficiency. Finally, a pulsatile flow is created via a cyclic clockwise twisting and releasing mechanism upon relatively fast deformation with the frequency of one cycle per second (Supplementary Video 3). Therefore, the torsion-induced contraction mechanism provides new design opportunities for converting a torsional (rotational) movement into pulsatile flow or squeezing a container containing yield stress liquids that are not easy to pump. The proposed applications could be of potential interest in soft robotics specifically when a limited cylindrical space is available which should be efficiently used. This system could obtain high precision on the flow rate and pressure by tuning the rotation angle and rotation rate.

In summary we showed that, particular orthotropy orientations in negative Poisson's ratio (auxetic) cylindrical shells can circumvent torsional buckling. Auxetic cylindrical shells with periodicities along helical paths trigger a negative radial strain when twisted and ultimately circumvent the torsional buckling under large torsional deformation. We revealed that contraction of the helical meta-shells under torsion coincides with softening followed by stiffening of the meta-shell, due respectively to a snap-through instability and ordered unit-cell stacking. These phenomena account for the cylinder's extreme resistance against buckling. The auxetic meta-shells display a sign-switching axial strain, from positive to negative, during the torsion, representing a transition from positive to reversed Poynting effect. We highlighted auxeticity and orthotropy orientation associated with a sign-switching Poynting response as two essential features in controlling the torsional buckling instability. As preventing torsional buckling is crucial in many mechanical systems functioning under torsion and compression, from robotic arms to biological systems like blood vessels, this study provides pathways to designing robust mechanical components for a wide range of applications. The radial contraction mechanism introduced here can offer a novel strategy for designing torsional compressors and valves, which potentially can be used to mimic the pumping of blood in the heart that happens through complex twisting-contraction mechanisms[50,51].

## Methods

### Design principles and parameters

Our auxetic metamaterial sheets consist of 2D arrays of holes of radius $r$. The holes are separated by so-called hinges spanning a distance $t$, as shown in Fig. 6. We show straight (Fig. 6a) and rotated (Fig. 6b) unit cells and their geometrical parameters. To realize this design in a cylindrical structure, circles were initially placed at the outer radius of the cylindrical shell and then extruded in the radial direction toward the main axis of the cylinder to create the void volumes (Fig. 6c, left). We mapped the void network on cylindrical coordinates to describe the auxetic cylindrical shells (Fig. 6c, right).

First, we created a symmetric cylinder with $n = 12$ 'straight' unit cells around the circumference and 8 unit cells over the height of a

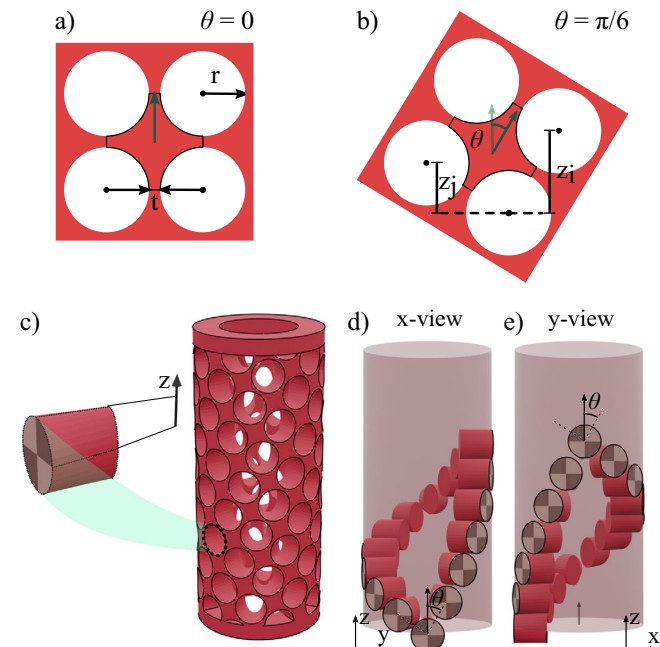

**Fig. 6 | Cylinder design and parameters. a** Contour of the unit cell, as created by creating circular voids in a sheet that form a diamond-like shape with a diagonal along the z-axis of the cylinder. **b** Same unit cell rotated by $\theta = \pi/6$, showing the positions of the voids in 2D coordinates. **c** Void shape in a cylindrical coordinate system, created by extruding circular voids towards the shell axis. **d, e** Helical meta-shell with $\theta = \pi/6$. The 6th void in clockwise and the 10th void in counterclockwise helical tiling fully overlap, shown from the x-view (**d**) and y-view (**e**).

cylinder with an initial inner and outer radii of $R_{0,\min} = 7.5$ mm and $R_{0,\max} = 12.5$ mm, respectively (Fig. 2d). The hinge thickness is set to $t = 0.6$ mm at the outer radius, and the initial height of the cylinder is $h_0 = 53.3$ mm. We clamp the structure using two disks with the same inner and outer radii and a height of 3 mm. Experimental data are based on one sample for each meta-shell design. However, to ensure that the results are reproducible, a limited number of additional samples (with $\theta = \pi/6$) were tested and compared qualitatively, confirming the reproducibility of the experimental results. Moreover, the FE results are in excellent agreement with experimental results, indicating the reliability of the experiments.

### Cylindrical boundary condition

To map the tilted unit-cell network on cylindrical coordinates, we considered the periodic boundary condition in the circumferential direction. In other words, unit cells with clockwise (cw) and counterclockwise (ccw) helical tiling must precisely overlap where they meet. If unit cell $n$ in the cw helix overlaps with unit cell $m$ in the ccw helix (counting from 0), the unit cell's rotation angle is given by $\theta = \arctan(n/m)$. Since $n$ and $m$ are integers, practically, we can only create rotated designs with specific rotation angles. We fabricated meta-shells with $\theta = \arctan(0/12) = 0$, $\theta = \arctan(3/12) \approx \pi/12$, $\theta = \arctan(5/11) \approx \pi/8$, $\theta = \arctan(6/10) \approx \pi/6$, $\theta = \arctan(7/9) \approx 5\pi/24$ and $\theta = \arctan(8/8) = \pi/4$.

A pattern of voids with given $(\varphi_i, z_i)$ coordinates for the cw and $(\varphi_j, z_j)$ coordinates for the ccw directions creates the helical network of unit cells on the cylindrical shells (Fig. 6b). The pitch values of the cw and ccw helices can be calculated by $2\pi R \cot(\theta)$ and $2\pi R \tan(\theta)$, respectively. Likewise, for these unit cells, corresponding void volumes overlap in helical arrays with opposite pitch values, an example of which is shown in Fig. 6d and e. This is satisfied by $m\varphi_m + n\varphi_n = 2\pi$, where $\varphi_n = \arccos[1 - \frac{(2r' + t)^2 \sin^2\theta}{2R_{\max}^2}]$ and $\varphi_m = \arccos[1 - \frac{(2r' + t)^2 \cos^2\theta}{2R_{\max}^2}]$. In

these equations, $t$ is the hinge thickness, and $R_{max}$ is the maximum radius of the meta-shell. The parameter $r'$ is the radius of the void circle, vertically placed at the outer shell surface while fully inside the meta-shell and touching it at the sides. However, for a precise design, we placed a circle touching the outer surface at the top and bottom (outlined in Fig. 6d and e), which is slightly bigger, given by $r = r' / \sqrt{1 - (r'/R_{max})^2}$. The values of $z_i$ and $z_j$ are calculated by $z_i = (2r + t)\cos\theta$ and $z_j = (2r + t)\sin\theta$, respectively. The discussed boundary condition imposes a limitation on the void size given by the radius of the outer circle, $r$, or thickness of the hinges, $t$ (the minimum gap between two neighboring voids). We kept the hinge thickness constant for all cylinders at $t = 0.6$ to assure consistency among different cylinders. To achieve this constant value, the radii of the outer contour of the voids were varied within the small range of $r = 3.1 \pm 0.2$ mm. STL files for 3D printing and 3D visualization of the designs were created using Blender, and STEP files for FE simulations using FreeCAD.

### Fabrication and experiments

We 3D printed the designed structures using a Formlab Form2 3D printer and elastic resin v1 with a 0.1 mm printing resolution. A bulk cylinder, 3D printed using the same machine and under the same conditions as the cylindrical metamaterials, has a Young's modulus of $Y = 2.7$ MPa. We applied the deformations using an Anton Paar 300 rheometer and measure the torque, normal force, torsion, and axial deformations. We did so at low strain rates of $\approx 0.25$ mm/min compression and $\approx 0.1$ rad/min torsion to obtain a quasi-static deformation process.

### Finite element simulations

A nonlinear Finite Element (FE) solver (Abaqus 2023.HF2, Dassault Systèmes Simulia Corp.) is used for FE simulations. Simulations are performed under the same conditions as the experiments. The 3D geometries of meta-shells with various hole configurations are directly imported into Abaqus. The gripping o-rings to clamp the top and bottom of the meta-shells are designed directly in Abaqus, and then integrated with the cylinder geometry using the "tie" option.

The Ogden hyperelastic material model (strain energy potential, $n = 1$) is used to describe the non-linear stress-strain behavior of the base material, as determined from uniaxial compression test on bulk and dogbone tensile test (For more information, read SI: Uniaxial experiments on the bulk samples). However, the elastic modulus of the FE results are calibrated according to the experimental effective Young's modulus in the range of small deformations ($-0.2$ rad $< \varphi < 0.2$ rad) of the meta-shells. Therefore, the torque responses are rescaled, where the scaling factor is the same for all samples, obtained as 2.06. A standard self-contact is defined using a surface-to-surface discretization method, incorporating tangential behavior with a penalty friction coefficient of 0.1, normal behavior with a "hard" contact pressure threshold, and allowing separation after contact. In order to apply the boundary conditions, two reference points are defined at the top and bottom centers of the gripping o-rings, which were attached to the cylinder. These reference points are kinematically coupled to the top and bottom surfaces of the gripping o-ring clams. The bottom reference point was fully constrained, while the top reference point had the freedom to move and rotate along and around the longitudinal axis of the cylinder. We applied a rotation of $\pi$ on the top node. The top reference point was constrained in the other directions. We employed quadratic tetrahedral elements (C3D10H, 10-node quadratic tetrahedron, hybrid, constant pressure) elements to mesh the geometries.

### Data availability

The data that support the findings of this study are available are available on the public repository https://doi.org/10.5281/zenodo.12800503

## Code availability

The codes that support the findings of this study are available on the public repository https://doi.org/10.5281/zenodo.12800503

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

## Acknowledgements
M. H. acknowledges funding from the Dutch Research Council NWO, through NWO-VIDI grant No. 680-47-548/983. C.C. acknowledges funding from the European Research Council under grant agreement 852587 and from the Netherlands Organisation for Scientific Research under grant agreement VI.Vidi.213.131.3. We are grateful to Mokhtar Adda-Bedia for fruitful discussions.

## Author contributions
A.G., C.C., D.B., E.L., and M.H. defined the research question and conceptualized and guided the research. A.G., C.C., D.B., and M.H. designed the experiments. A.G. designed the 3D models and conducted the experiments. A.G. performed the data acquisition, analysis, visualizations, and preparing the supplementary materials. T.R. performed the preliminary experiments and contributed to the proof of concept. M.J.M. performed the finite element simulations, data collection from the simulations, and visualizations of the finite element components. All authors contributed to the interpretation of the results and analysis methods. A.G. wrote the original draft. A.G., C.C., and M.H. revised the original draft. All authors critically revised the manuscript for its intellectual content and approved the manuscript.

## Competing interests
The authors declare no competing interests.
