## [Peer Review File · Nature Communications]

Suppressing torsional buckling in auxetic meta-shellsREVIEWER COMMENTS

Reviewer #1 (Remarks to the Author):

Review of "Suppressing torsional buckling in auxetic meta-shells" (NCOMMS-23-63609)

In this manuscript, the authors study how changing the unit cell orientation of cylindrical shells based on auxetic "holey sheets" affects the response to torsional loading. For some orientations, torsional buckling is suppressed in favor of radial contraction. Interesting behaviors emerge, including a sign-switching Poynting effect, high magnitude of the negative shear-induced contraction ratio, and different folding of unit cells upon one another compared to a similar metamaterial under compressive loading. I think the experiments are well-done and the analysis is interesting, so this work may merit publication in Nature Communications. However, I have several concerns that should be addressed first. Thus, I recommend major revisions.

MAJOR COMMENTS:

1. The writing often distracts from the message of the work and needs to be improved. I strongly recommend edits by a native English speaker to improve clarity. For instance, the following sentence is unclear (p. 2, lines 56-62: "In isotropic systems, shear deformation usually leads to a perpendicular dilation, typical Poynting effect [15], or contraction, rare reversed Poynting effect [16], but an anisotropic system can exhibit a sign-switching Poynting response [17].") There are similar examples throughout the manuscript, and some vocabulary is slightly confusing. Furthermore, the introduction is not concise, particularly the last paragraphs beginning with line 51. I suggest revisions for clarity.

2. In the introduction, lines 63-75, the authors situate their work somewhat within the landscape of auxetic mechanical metamaterials. However, I do not feel that the authors distinguish their work from related studies in lines 66-68 and 71-73: "... cylindrical [28-32] metamaterials with various unique functionalities that mainly rely on the internal buckling of their building blocks.", and "...porosity-[dependent] buckling in holey cylindrical shells [29].". Some clarification of what has been done in the existing literature would be helpful for readers to assess novelty, especially for studies which employ torsional loading like [28].

MINOR COMMENTS:

3. Page 3, line 112: "As a result of this modification, the horizontal and vertical lines along which the unit-cells are connected, in the direction of principal axes, change to helices with opposite handedness (growing with opposite rotations) and different pitch values on the cylindrical shell." This sentence is hard to understand.

4. Page 3, line 114: It may be worth defining "pitch values" in this context.

5. Page 5, line 198: "This folding mechanism... leads to a negative radial strain due to compaction and filling of the voids." Is this actually different from the well-known case [18]?

6. Section 3.3. How does this torsional compressor compare to other designs/pumps that drive pulsatile flow? Are there particular advantages?

7. Page 11, Figure 1b: it is difficult to see the deformation (especially the indented region).

8. Page 12, Figure 2f-g: What does the [-] signify on the axes? If it's indicating that the axes are dimensionless, I think this is unnecessary and possibly confusing. The same goes for Fig 3b.

Reviewer #2 (Remarks to the Author):

It is shown that torsional buckling in cylindrical shells can be eliminated through a clever design of a "meta-shell", involving both auxeticity and orthotropy orientation. Finally a proof-of-concept

prototype for a soft torsional compressor is presented.

Related to the mechanical behaviour and failure of several man-made and natural structures, torsional instabilities still represent an open research topic and challenging scientific subject.

The article is topical, original, clearly written, and the supporting material is very well organized. I have no hesitation in recommending acceptance of the paper for publication in Nature Communications.

Reviewer #3 (Remarks to the Author):

The paper presents an experimental investigation on the suppression of the torsional buckling of cylindrical shells, if these present a 'holey sheets' pattern, thus being auxetic.

To my knowledge, this is the first time this experiment is performed and characterized, with the authors performing several experiments in terms of the chirality of the pattern, its orientation with respect to the vertical axis, and the direction of torsion (counter clockwise or clockwise).

The results show that, when the principal axes of the orthotropic meta-shells are relatively aligned with that of the compressive component of the applied stress during twisting, the so-called meta-shell uniformly shrinks in the radial direction.

While the study is new and the experimental procedure is sound, there is neither a theoretical model nor finite element simulations that support the findings.

For example, a finite-element campaign could elucidate some additional features of the phenomenon, while also providing a fast tool for design. A theoretical model, on the other side, could explain in more details while the phenomenon occurs. As the material is auxetic only along the mid-surface of the shell, the dimensional reduction procedure should take into account these different constitutive laws between the in-plane and the out-of-plane directions.

My belief is that, for the paper to be considered to be published in Nature Communications, it needs at least one of the two additional components I mentioned, that is a theoretical model and/or a finite element simulation campaign.

Reviewer #4 (Remarks to the Author):

Manuscript#: NCOMMS-23-63609

- Key results: Please summarize what you consider to be the outstanding features of the work.
 - o Prevent torsional buckling instability by combining auxeticity and tuning orthotropy. Torsional instability was prevented by radial contraction due to local buckling. A potential application for creating pulsatile flow by selective radial contraction depends on the direction of applied torsion

- Validity: Does the manuscript have flaws which should prohibit its publication? If so, please provide details
 - o No comments

- Originality and significance: If the conclusions are not original, please provide relevant references. On a more subjective note, do you feel that the results presented are of immediate interest to many people in your own discipline, and/or to people from several disciplines?
 - o I do feel the results presented are of immediate interest to the readers in the field of mechanics, soft robotics, metamaterials, and many more. This is a nice addition to the existing literature on utilizing the instabilities for functionality.

•Data & methodology: Please comment on the validity of the approach, quality of the data and quality of presentation. Please note that we expect our reviewers to review all data, including any extended data and supplementary information. Is the reporting of data and methodology sufficiently detailed and transparent to enable reproducing the results?

o The methodology followed in this paper is sound.

o In line 154, the radial strain equation should be $e_r = (r - R_{\max}) / R_{\max}$ to avoid any confusion.

•Appropriate use of statistics and treatment of uncertainties: All error bars should be defined in the corresponding figure legends; please comment if that's not the case. Please include in your report a specific comment on the appropriateness of any statistical tests, and the accuracy of the description of any error bars and probability values.

o Throughout the paper, it's unclear how many samples were tested for each experiment, and the error estimations are missing. However, this is acceptable since the goal is to demonstrate the global trends and behaviors of the considered helical metastructure. Even though proving the observed behavior's consistency is beneficial and strengthens the derived conclusions.

•Conclusions: Do you find that the conclusions and data interpretation are robust, valid and reliable?

o Yes

•Suggested improvements: Please list additional experiments or data that could help strengthening the work in a revision.

o No comment

•References: Does this manuscript reference previous literature appropriately? If not, what references should be included or excluded?

o No comment

•Clarity and context: Is the abstract clear, accessible? Are abstract, introduction and conclusions appropriate?

o Yes, the clarity and context of the manuscript is appropriate

•Inflammatory material: Does the manuscript contain any language that is inappropriate or potentially libelous?

o No.

•Springer Nature is committed to diversity, equity and inclusion; please raise any concerns that may in your view have an impact on this commitment.

o No comments

•Please indicate any particular part of the manuscript, data, or analyses that you feel is outside the scope of your expertise, or that you were unable to assess fully.

o No comments

Reply to reviewers:

We thank the reviewers for their careful reading of the manuscript and constructive comments.

Reviewer #1

In this manuscript, the authors study how changing the unit cell orientation of cylindrical shells based on auxetic "holey sheets" affects the response to torsional loading. For some orientations, torsional buckling is suppressed in favor of radial contraction. Interesting behaviors emerge, including a sign-switching Poynting effect, high magnitude of the negative shear-induced contraction ratio, and different folding of unit cells upon one another compared to a similar metamaterial under compressive loading. I think the experiments are well-done and the analysis is interesting, so this work may merit publication in Nature Communications. However, I have several concerns that should be addressed first. Thus, I recommend major revisions.

1. The writing often distracts from the message of the work and needs to be improved. I strongly recommend edits by a native English speaker to improve clarity. For instance, the following sentence is unclear (p. 2, lines 56-62: "In isotropic systems, shear deformation usually leads to a perpendicular dilation, typical Poynting effect [15], or contraction, rare reversed Poynting effect [16], but an anisotropic system can exhibit a sign-switching Poynting response [17].") There are similar examples throughout the manuscript, and some vocabulary is slightly confusing. Furthermore, the introduction is not concise, particularly the last paragraphs beginning with line 51. I suggest revisions for clarity.

We apologize for the unclarity. We have rephrased the mentioned sentences in the revised version of the manuscript to improve clarity. The new version of the manuscript is edited by a native English speaker and revisions are made in different parts. We think the new version is written clearly.

2. In the introduction, lines 63-75, the authors situate their work somewhat within the landscape of auxetic mechanical metamaterials. However, I do not feel that the authors distinguish their work from related studies in lines 66-68 and 71-73: "... cylindrical [28-32] metamaterials with various unique functionalities that mainly rely on the internal buckling of their building blocks.", and "...porosity-[dependent] buckling in holey cylindrical shells [29]." Some clarification of what has been done in the existing literature would be helpful for readers to assess novelty, especially for studies which employ torsional loading like [28].

In the revised version of the manuscript, we have addressed the concern of the reviewer in the introduction section by clarifying the scope of previous research and providing a more comprehensive picture of the previous studies. We have also clarified our aim and findings to distinguish the novelty of our work. We mentioned that the torsional loading of metamaterials is rather poorly explored. The pioneering work of Lazarus and Reis (2015) was focused on actuating the system via decreasing the inner pressure of the shell, which in some cases led to torsional reformations but direct torsional loading

is not explored. However, we referred to our previous research, investigating the axial response in a different cylindrical metamaterial under torsion deformations with a completely new aim and new results.

3. Page 3, line 112: "As a result of this modification, the horizontal and vertical lines along which the unit-cells are connected, in the direction of principal axes, change to helices with opposite handedness (growing with opposite rotations) and different pitch values on the cylindrical shell." This sentence is hard to understand.

In the revised version of the manuscript, we rephrased these sentences to improve clarity.

4. Page 3, line 114: It may be worth defining "pitch values" in this context.

We defined the pitch values in the "Methods" section of the revised version.

5. Page 5, line 198: "This folding mechanism... leads to a negative radial strain due to compaction and filling of the voids." Is this actually different from the well-known case [18]?

In the well-known case mentioned by the reviewer, the rotation of unit cells is opposite and unit cells along the principal axes fold on each other. In the case of twisting meta-shells (our system), however, all unit cells rotate in the same direction and fold on each other. In Figure 2e, in the revised version we have addressed the differences in folding mechanisms between the two cases.

6. Section 3.3. How does this torsional compressor compare to other designs/pumps that drive pulsatile flow? Are there particular advantages?

The system that we introduced provides new design opportunities for converting a torsional (rotational) movement into pulsatile flow or squeezing a container containing yield stress liquids that are not easy to pump. This could be of potential interest in soft robotics specifically when a limited space/cylindrical space is available which should be efficiently used. The system could obtain high precision on the flow rate and pressure by tuning the rotation angle and rotation rate.

7. Page 11, Figure 1b: it is difficult to see the deformation (especially the indented region).

We replaced Figure 1b with a picture that shows the deformation more clearly.

8. Page 12, Figure 2f-g: What does the [-] signify on the axes? If it's indicating that the axes are dimensionless, I think this is unnecessary and possibly confusing. The same goes for Fig 3b.

We agree with the reviewer and we made the requested changes to Figures 2–4 in the revised version to implement this comment.

Reviewer #2

It is shown that torsional buckling in cylindrical shells can be eliminated through a clever design of a "meta-shell", involving both auxeticity and orthotropy orientation. Finally a proof-of-concept prototype for a soft torsional compressor is presented.

Related to the mechanical behaviour and failure of several man-made and natural structures, torsional instabilities still represent an open research topic and challenging scientific subject.

The article is topical, original, clearly written, and the supporting material is very well organized.

I have no hesitation in recommending acceptance of the paper for publication in Nature Communications.

We thank the reviewer for their positive evaluation of our work.

Reviewer #3

The paper presents an experimental investigation on the suppression of the torsional buckling of cylindrical shells, if these present a 'holey sheets' pattern, thus being auxetic.

To my knowledge, this is the first time this experiment is performed and characterized, with the authors performing several experiments in terms of the chirality of the pattern, its orientation with respect to the vertical axis, and the direction of torsion (counter clockwise or clockwise).

The results show that, when the principal axes of the orthotropic meta-shells are relatively aligned with that of the compressive component of the applied stress during twisting, the so-called meta-shell uniformly shrinks in the radial direction.

While the study is new and the experimental procedure is sound, there is neither a theoretical model nor finite element simulations that support the findings.

For example, a finite-element campaign could elucidate some additional features of the phenomenon, while also providing a fast tool for design. A theoretical model, on the other side, could explain in more details while the phenomenon occurs. As the material is auxetic only along the mid-surface of the shell, the dimensional reduction procedure should take into account these different constitutive laws between the in-plane and the out-of-plane directions.

My belief is that, for the paper to be considered to be published in Nature Communications, it needs at least one of the two additional components I mentioned, that is a theoretical model and/or a finite element simulation campaign.

We thank the reviewer for his/her constructive feedback. We agree with the reviewer, therefore, we performed a thorough finite-element analysis of the system. We successfully confirmed the experimental results showing torsion-induced radial contraction depending on the orthotropy of the meta-shells. We investigated additional samples with different unit-cell orientations (FEA: $\theta=0^\circ, 14^\circ, 24^\circ, 31^\circ, 38^\circ, 45^\circ$) to obtain a parametric analysis on the dependency of contraction ratio on θ (Figures 1f–k). We better clarified the folding mechanism and softening/stiffening behaviors using FEA represented in Figure 4. Additionally, by varying the thickness of the meta-shell ($\theta=31^\circ$), we investigated the critical shell thickness to obtain radial contraction instead of buckling, which is presented in the supplementary information. More details are provided in the edited/new text in the manuscript. All the major edits are highlighted in red in the revised version of the manuscript. The FEA not only helps to better understand the underlying mechanics and visualize them, it also provides a rapid tool for tailor-made design of such systems for potential applications.

Reviewer #4

We acknowledge the reviewer for his/her positive evaluation and constructive feedback.

In line 154, the radial strain equation should be $e_r = (r - R_{\max}) / R_{\max}$ to avoid any confusion. In the revised version of the manuscript, we have changed the equation as requested.

Throughout the paper, it's unclear how many samples were tested for each experiment, and the error estimations are missing. However, this is acceptable since the goal is to demonstrate the global trends and behaviors of the considered helical metastructure. Even though proving the observed behavior's consistency is beneficial and strengthens the derived conclusions.

In the revised version of the manuscript, we included the additional information in the Methods section to clarify the experimental results are based on one sample for each case. However, additional samples were tested and compared qualitatively, confirming the reproducibility of the experiments. Moreover, newly added finite element results are in excellent agreement with the experimental results, which confirm that the experimental procedures are reliable.

REVIEWERS' COMMENTS

Reviewer #1 (Remarks to the Author):

In the authors' revised manuscript, the clarity and figures are much improved. I feel that the Finite Element Analysis adds to the work, and some of my comments have been addressed. However, I have remaining comments that I believe should be addressed before this manuscript is accepted for publication.

1. In the introduction (p. 2, lines 59-70), the authors now better explain the findings of some of the previous literature, which helps to distinguish the novelty of the present work. I appreciate the clarification in the authors' response of the distinction between the current work and that of Lazarus and Reis (2015) (i.e. that this study employs torsional loading, whereas Lazarus and Reis decreased internal pressure.) However, this reference (Lazarus and Reis; [28] in the original manuscript) and Zhao, et al., "Three-Dimensionally Printed Mechanical Metamaterials With Thermally Tunable Auxetic Behavior", *Physical Review Applied*, 2019 ([27] in the original manuscript) are inexplicably missing from the revised version. Perhaps this was an oversight by the authors, but these omissions are inappropriate -- all references should be checked carefully and the aforementioned works, especially the closely related work of Lazarus and Reis, should be credited in this study.

2. In response 6 the authors have answered my question about the advantages of the application discussed in section 3.3, but made no related change to the manuscript. As a result, the need for such a design of a torsional compressor/pumps is really not motivated. Simply adding a concise sentence about this would help.

3. The authors find terrific agreement between experiments and simulations in general, but not in Figure 3d. Why is torque overpredicted by FEA? Please address in the manuscript.

4. What is the source of the oscillations from experiments in fig 4d that are not present in simulations? Does this indicate friction in expts? Or do simulations miss some local instabilities?

Reviewer #3 (Remarks to the Author):

I am satisfied with the changes made. Specifically, the Authors have performed FE simulations and studied the effect of thickness.

This gives more robustness to the study.

Reviewer #4 (Remarks to the Author):

The authors have incorporated my suggestions and made the necessary revisions. I recommend this manuscript for publication, as it will make a valuable contribution to the field.

Reply to reviewer one:

We thank the reviewers for their positive evaluation of our work. Below we have addressed the comments raised by the first reviewer.

In the authors' revised manuscript, the clarity and figures are much improved. I feel that the Finite Element Analysis adds to the work, and some of my comments have been addressed. However, I have remaining comments that I believe should be addressed before this manuscript is accepted for publication.

We thank the reviewer for the positive evaluation of our work. Below we have addressed the comments in detail.

1. In the introduction (p. 2, lines 59-70), the authors now better explain the findings of some of the previous literature, which helps to distinguish the novelty of the present work. I appreciate the clarification in the authors' response of the distinction between the current work and that of Lazarus and Reis (2015) (i.e. that this study employs torsional loading, whereas Lazarus and Reis decreased internal pressure.) However, this reference (Lazarus and Reis; [28] in the original manuscript) and Zhao, et al., "Three-Dimensionally Printed Mechanical Metamaterials With Thermally Tunable Auxetic Behavior", Physical Review Applied, 2019 ([27] in the original manuscript) are inexplicably missing from the revised version. Perhaps this was an oversight by the authors, but these omissions are inappropriate -- all references should be checked carefully and the aforementioned works, especially the closely related work of Lazarus and Reis, should be credited in this study.

We apologize for the accidental omission of the important references in the revised version. We thank the reviewer for careful checking the references. In the revised version of the manuscript, we have added those references, listed as reference 24 and 30 in the revised version.

2. In response 6 the authors have answered my question about the advantages of the application discussed in section 3.3, but made no related change to the manuscript. As a result, the need for such a design of a torsional compressor/pumps is really not motivated. Simply adding a concise sentence about this would help.

In the revised version of the manuscript, we have added the following sentences to motivate the application of this system in the novel design of the torsional compressor.

“Here we propose exploiting torsion-induced localized or pulsatile contractions for designing pumps and compressors.

...

Therefore, the Torsion-induced contraction mechanism provides new design opportunities for converting a torsional (rotational) movement into pulsatile flow or squeezing a container containing yield stress liquids that are not easy to pump. The proposed applications could be of potential interest in soft robotics specifically when a limited cylindrical space is available which should be efficiently used. This system could obtain high precision on the flow rate and pressure by tuning the rotation angle and rotation rate.”

3. The authors find terrific agreement between experiments and simulations in general, but not in Figure 3d. Why is torque overpredicted by FEA? Please address in the manuscript.

We thank the reviewer for raising this point. We have discussed the discrepancy between the FE and experimental results in section 3.2. The largest deviation between the experimental and FE results is about 19 %. The source of deviation often stems from inaccuracies in the determination of the material model under extreme deformation conditions. An additional factor that may cause deviation is rooted in differences between the contact properties (e.g. friction coefficient) of the real system and the FE simulation. To address them, we have revised the paragraph and added a few more sentences in the revised version of the manuscript as follows.

“The FE results predict similar values for the onset of buckling in systems that undergo buckling. The buckling torsional angles predicted by FEA are in excellent agreement with the experimental results (Figure 6c). However, the buckling torques derived from the FE results exhibit deviations from the experimental data (Figure 6d). Nonetheless, this deviation remains within an acceptable range, with the largest deviation being only 19%, observed for the meta-shell with $-\theta = \pi/12$. This discrepancy primarily stems from inaccuracies in the material model under extreme deformation conditions. Moreover, our assumption in defining the contact properties in FE simulations (e.g., friction coefficient of 0.1) may differ from the real system, which could result in deviations from the obtained buckling torque values in the experiments”.

4. What is the source of the oscillations from experiments in fig 4d that are not present in simulations? Does this indicate friction in exps? Or do simulations miss some local instabilities?

We thank the reviewer for raising this point. The fluctuations which are not observed in the FE results are due to experimental inaccuracies in determining the torque. We added the following sentence in the revised version of the manuscript.

“Fluctuations in the experimental local stiffness values arise from experimental inaccuracies in the torque measurements, which are not relevant for the FE results (See Supplementary Information, Local stiffness calculation).”

The following paragraph is also added to Supplementary Information.

“Local stiffness calculation

The stiffness modulus is given by $g_s = (h/J)(d\tau / d\varphi)$, where $d\tau / d\varphi$ is the differential of the torque with respect to torsion, and calculated numerically from the experimental data. Very small $d\varphi$ steps may produce large fluctuations in local stiffness due to experimental inaccuracy in the torque response. Therefore, to mitigate the fluctuations we increased the step size of the torsional deflection to $d\varphi=110\text{si}\{\text{mrad}\}$ and kept enough data points.”